# Obesity in Young Adulthood: The Role of Physical Activity Level, Musculoskeletal Pain, and Psychological Distress in Adolescence (The HUNT-Study)

**DOI:** 10.3390/ijerph17124603

**Published:** 2020-06-26

**Authors:** Maren Hjelle Guddal, Synne Øien Stensland, Milada Cvancarova Småstuen, Marianne Bakke Johnsen, Ingrid Heuch, John-Anker Zwart, Kjersti Storheim

**Affiliations:** 1Research and Communication Unit for Musculoskeletal Health (FORMI), Oslo University Hospital, P.O. Box 4956 Nydalen, 0424 Oslo, Norway; synne.stensland@nkvts.no (S.Ø.S.); m.b.johnsen@medisin.uio.no (M.B.J.); kjersti.storheim@medisin.uio.no (K.S.); 2Faculty of Medicine, Institute of Clinical Medicine, University of Oslo, P.O. Box 1078 Blindern, 0316 Oslo, Norway; j.a.zwart@medisin.uio.no; 3Norwegian Centre for Violence and Traumatic Stress Studies, P.O. Box 181 Nydalen, 0409 Oslo, Norway; 4Faculty of Health Sciences, Oslo Metropolitan University, P.O. Box 4, St. Olavs plass, 0130 Oslo, Norway; miladacv@medisin.uio.no; 5Department of Research, Innovation and Education, Division of Clinical Neuroscience, Oslo University Hospital, Oslo, Ullevål, P.O. Box 4956 Nydalen, 0424 Oslo, Norway; UXINHE@ous-hf.no

**Keywords:** obesity, prevention, adolescence, young adulthood, physical activity, musculoskeletal pain

## Abstract

The global obesity epidemic raises long-term health concerns which underline the importance of preventive efforts. We aimed to investigate individual and combined effects of common health problems in adolescence on the probability of obesity in young adulthood. This prospective population-based study included data from participants in the Nord-Trøndelag Health Study in Norway (Young-HUNT1 (1995–1997), age 13–19, baseline) who participated in HUNT3 as young adults 11 years later (age 23–31). Exposure variables at baseline included self-reported physical activity, musculoskeletal pain, and psychological distress. We examined associations between exposure variables and the main outcome of obesity in young adulthood (BMI ≥ 30 kg/m^2^) using univariate and multiple logistic regression, stratified by sex. Probabilities of obesity for given combinations of the exposure variables were visualized in risk matrixes. The study sample consisted of 1859 participants (43.6% boys). Higher probabilities of obesity in young adulthood were found across combinations of lower physical activity levels and presence of musculoskeletal pain in adolescence. Additional adverse effects of psychological distress were low. Proactive intervention strategies to promote physical activity and facilitate sports participation for all adolescents, whilst addressing musculoskeletal pain and its potential individual causes, could prove helpful to prevent development of obesity in young adulthood.

## 1. Introduction

Obesity is a major public health concern worldwide, and is associated with increased risk of chronic morbidity and mortality [1]. A substantial rise in the prevalence of obesity during the last three decades [1,2], along with an increase in related comorbidities and disease burden [2], has made the prevention of obesity a global public health priority [3]. Adolescence, and the subsequent transition into young adulthood, have been recognized as critical periods for weight-related behavior change to prevent the development of obesity [4,5]. 

A lifestyle characterized by low levels of physical activity (PA), musculoskeletal pain, and psychological distress are three common and related factors adversely affecting the health and well-being of adolescents worldwide [6,7,8,9,10]. The cumulative presence of these factors could be of particular importance for the development of overweight and obesity over time. Particularly, regular PA is considered one of the most important lifestyle factors in the prevention and management of obesity [11] and long-term morbidity [12]. Young people commonly drift away from engagement in regular physical activity throughout their adolescent years [13], and the majority of adolescents worldwide (81%) do not adhere to the current recommendations of daily PA [6]. Emerging unhealthy adolescent behavior patterns, such as no or infrequent engagement in PA, could affect the development of obesity into adulthood [14,15]. However, there is a lack of knowledge about the impact of early regular PA on later risk of obesity. More specifically, results from systematic reviews are inconsistent and have provided limited evidence for the contribution of PA in adolescence on development of obesity [14,15,16], with few studies covering the transition period from adolescence to young adulthood. 

Further, musculoskeletal pain and mental health problems frequently take place in adolescence, and are among the top causes of functional impairments [7,8,10,17]. Estimations indicate that between one third and one half of adolescents worldwide report musculoskeletal pain monthly or more [7], and up to one in five children and adolescents experience mental health problems [8,18]. Moreover, these problems commonly co-occur [19,20]. In a national representative cohort from the United States, 26% of adolescents reported that they had experienced both chronic pain and mental health problems during their lifetime [21]. Both musculoskeletal pain and mental health problems are known to be leading causes of health-related disability [8,10,22], associated with impairments in daily activities, in social relationships and in extracurricular activities, as well as sleep disturbances and absence from school [7,19,23]. 

There is a possibility that these commonly persistent health problems, alongside lack of PA, could impact heavily on young people’s capability to pursue healthy lifestyles. The potential adverse impact of pain on the development of obesity has been raised as a concern based on the high prevalence of obesity seen in adolescents with chronic pain [24,25]. Further, results from systematic reviews have found psychological distress or depression/depressive symptoms in adolescence being predictive for development of obesity [26,27,28,29]. Limited evidence, however, exists on individual and combined effects of these common health problems from population-based cohorts with follow-up into young adulthood. 

Young adults of this generation appear to be particularly affected by the global obesity epidemic [30], highlighting the importance of addressing the weight-related consequences of the potential risk factors. However, the question of whether the co-occurrence of physical inactivity and these health problems in adolescence may play a role in the development of obesity has received little attention. As current preventive efforts to reduce obesity seem to be insufficient [31], there is need for further exploration of factors that may be helpful in guiding intervention strategies. Particularly, future prospective studies assessing the combination of multiple potential risk factors for obesity are warranted [32]. Early intervention is a key in the prevention of obesity [31,33,34], and knowledge about the potential impact of the co-existence of health challenges in adolescence may make a valuable contribution to this initiative. Physical activity habits, as well as musculoskeletal pain and mental health challenges, are likely to be less ingrained in adolescence than in adulthood, emphasizing the importance of preventive efforts during the adolescent years when these potential risk factors are more malleable [35]. Identifying groups of adolescents potentially at risk for the development of obesity could help to provide early, targeted preventive interventions. 

To the best of our knowledge, it has not been examined in large population-based samples whether the joint impact of physical inactivity, musculoskeletal pain, and psychological distress in adolescence increases the probability of obesity. Thus, the aim of this prospective study with 11-year follow-up was to examine the impact of the individual and combined occurrence of PA levels, musculoskeletal pain, and psychological distress in adolescence on the probability of obesity in young adulthood. We hypothesized that physical inactivity, musculoskeletal pain, and psychological distress in adolescence would contribute additively to adverse impact on the probability of obesity in young adulthood.

## 2. Materials and Methods 

### 2.1. Study Sample

All adolescents aged 13–19 years living in the county of Nord-Trøndelag in Norway (*n* = 10,202) were invited to participate in the first adolescent phase of the Nord-Trøndelag Health Study, the Young-HUNT1 study, conducted in 1995–1997. During school hours, the adolescents completed a comprehensive health-related questionnaire about lifestyle factors, health, and quality of life, including the exposure variables described below (Section 2.3 Exposures). In addition, nurses performed health examinations that included measurement of height and weight at school visits. In total, 8983 adolescents responded the questionnaire in Young-HUNT1 (response rate 88%). These adolescents were also invited as young adults 11 years later, in 2006–2008, to participate in the HUNT3 study. Of the invited young adults who had participated in the Young-HUNT1 (*n* = 5353), 36% participated in the 11-year follow-up in HUNT3 [36]. Participants in HUNT3 completed a comprehensive health-related questionnaire at home, and attended screening stations for clinical examinations, including measurement of weight and height. In total, 1919 individuals participated in the Young-HUNT1 (baseline) as adolescents and in HUNT3 as young adults (age 23–31) 11 years later. We excluded participants ≥20 years of age at baseline (*n* = 10) according to the World Health Organization (WHO)’s definition of adults, and participants in HUNT3 who did not attend the clinical examination (*n* = 11) or did not have available BMI measures (*n* = 10). Those classified as underweight in HUNT3 (BMI < 18.5) were also excluded (*n* = 29) as these individuals are considered at higher risks of serious illness, leaving N = 1859 for the analysis (Appendix A). 

The majority of the population in the Nord-Trøndelag County is Caucasian. In terms of sex, age distribution, mortality and health status, the population is representative of Norway [36,37], and the development of obesity in Nord-Trøndelag is comparable to the global trend [1]. The Norwegian society is a socioeconomically stable society with a high degree of equity among students and young people [38]. A more comprehensive description of the data collection in the HUNT studies can be found elsewhere [36,37]. Participation in the study was voluntary. Inclusion was based on written consent from participants 16 years of age or older, and from the parents of those under 16 years of age, in accordance with Norwegian law. The Regional Committee for Medical Research Ethics (REK) has approved the current study (2014/1228/REK Sør-Øst A). The HUNT Studies have been approved by REK and the Norwegian Data Inspectorate of Norway. The Strengthening the Reporting of Observational studies in Epidemiology (STROBE) guidelines were used to ensure the reporting of this observational study.

### 2.2. Variables

Data on gender and age were obtained from the Norwegian National Population Registry. BMI was calculated by dividing body weight in kilograms by the squared value of body height in meters (kg/m^2^). The BMI-derived categorization of obesity for adolescents at baseline was defined in accordance with the extended International Obesity Task Force (IOTF) classification according to age and sex specific cut-off points [39].

### 2.3. Exposures

#### 2.3.1. Physical Activity Level (Young-HUNT1)

A validated question on frequency of PA from the World Health Organization Health Behavior in Schoolchildren (WHO HBSC) Survey Questionnaire was used to assess leisure time PA level [40]: “Outside school hours, how often do you usually exercise in your free time so much that you get out of breath or sweat?”. Response options were: every day, 4–6 days/week, 2–3 days/week, 1 day/week, less than every week, less than every month, and never. These seven categories were combined into three levels: ‘‘Low PA’’ (≤1 days/week) (reference group), ‘‘moderate PA’’ (2–3 days/week), and ‘‘high PA’’ (≥4 days/week).

#### 2.3.2. Musculoskeletal Pain (Young-HUNT1)

Musculoskeletal pain was assessed by questions regarding occurrence of neck or shoulder pain and joint or muscle pain in the past 12 months. Response options for frequency of pain were: “never”, “seldom”, “sometimes”, and “often”. Those who reported experiencing pain “sometimes” or “often” were classified as having musculoskeletal pain.

#### 2.3.3. Psychological Distress (Young-HUNT1)

A validated short version of the Hopkins Symptom Checklist Five items (SCL-5) was used to assess symptoms of psychological distress [41]. The statements included in SCL-5 were: “During the last 14 days: I have been constantly afraid and anxious; I have felt tense or uneasy; I have felt hopeless about the future; I have felt dejected or sad; I have worried too much about various things”. The four response options ranged from 1 = “not at all bothered” to 4 = “extremely bothered”. A mean score was calculated, with a cut-off for symptoms of psychological distress set at a mean score of two [41].

### 2.4. Outcome

#### Obesity (BMI ≥ 30 kg/m^2^) in Young Adulthood (at 11 Years Follow-Up, HUNT3)

Height and weight were measured by nurses. Height was measured to the nearest centimeter, and weight to the nearest half kilogram with light clothes, without shoes, jacket, or outdoor clothing. BMI was calculated (kg/m^2^), and those with BMI ≥ 30 were categorized as obese according to the definition of adult obesity adopted from the WHO.

### 2.5. Data Analyses

Categorical variables were reported as counts and percentages, and continuous variables as means and standard deviations (SDs). Analyses were stratified by gender. Differences in distribution of exposure variables at baseline between girls and boys were assessed using the Chi-square test for categorical variables and the Student’s t-test for continuous variables. 

The risk matrixes were constructed by first performing univariate logistic regression analyses to estimate the odds for the main outcome of obesity for each of the exposure variables; PA level (low, moderate, high), musculoskeletal pain (yes/no), and psychological distress (yes/no). Reference groups in the logistic regression analyses were young adults with BMI 18.5–29.9 kg/m^2^. The results were expressed as crude and adjusted odds ratios (ORs) with 95% confidence intervals (CIs). 

In our study, the aim was to develop a prediction model, not to reveal variables which would be independently associated with the outcome. Some variables might be statistically significant only when other variables are included in the multiple model, so we did not want to rule out such possible interactions. We therefore chose to include variables from the univariate analyses with *p*-values ≤ 0.2 into a multiple logistic regression model in order to evaluate whether their inclusion improved the overall model fit and the prediction abilities of the model. Thus, PA level and musculoskeletal pain for both sexes, in addition to psychological distress for girls, were included in multiple logistic regression models which formed the basis for calculations of the risk matrixes. The coefficients derived from the multiple models were used to compute probabilities for obesity in young adulthood given different combinations of the exposure variables at baseline. Risk matrixes for girls and boys were constructed separately to visualize the results. Sensitivity analyses were performed by limiting the reference group to participants within a normal weight range in young adulthood (BMI 18.5–24.9 kg/m^2^), excluding overweight participants (BMI 25–30 kg/m^2^). Statistical differences in exposure variables at baseline between Young-HUNT1 participants who also participated at HUNT3 (11-year follow-up) and those who did not participate in HUNT3 were calculated with the Chi square test. All analyses were considered exploratory so no correction for multiple testing was done and *p*-values < 0.05 were considered statistically significant. All tests were two-sided. Statistical analyses were performed with SPSS version 25 (SPSS Inc., Chicago, IL, USA).

## 3. Results

### 3.1. Characteristics of the Study Sample

The study sample consisted of 1859 participants, of whom 810 (43.6%) were males. Characteristics of the study participants at baseline (Young-HUNT1) by gender are presented in Table 1. Mean age was 16 years (SD 1.8). About one third of the adolescents reported a low level of PA (≤1 day/week). Musculoskeletal pain was frequently reported among both girls (36%) and boys (26%). The rate of psychological distress was twice as high in girls compared to boys (Table 1).

### 3.2. Probability of Obesity in Young Adulthood

In total, 165 (15.7%) girls and 146 (18.0%) boys were obese at 11 years follow-up (young adulthood, HUNT3). Results from univariate and multiple logistic regression analyses of PA level, musculoskeletal pain, and psychological distress in adolescence assessing potential predictors of obesity in young adulthood are listed in Table 2. A high PA level was the one of the selected covariates that was most strongly associated with reduced odds for obesity for both sexes.

Adolescent girls with a high level of PA, and no musculoskeletal pain or psychological distress, had the lowest probability of developing obesity as young adults (11%, 95% CI (6% to 16%)) (Table 3). The probability of obesity among girls was the highest among those with low PA in combination with musculoskeletal pain and psychological distress (25%, 95% CI (11% to 39%)). The risk matrix for boys, including PA level and musculoskeletal pain, showed similar results. Boys with a high PA level and no musculoskeletal pain had the lowest probability of obesity in young adulthood with 14% (95% CI (9% to 19%)), compared to 27% (95% CI (16% to 38%)) in boys with low PA who reported musculoskeletal pain (Table 4). Overall, about one in four adolescents with a low PA level in combination with musculoskeletal pain had developed obesity by young adulthood, compared to one in seven to nine adolescents with a high PA level and absence of musculoskeletal pain. 

Results from sensitivity analyses, with normal weight, rather than both normal and overweight individuals, as the reference group, suggested stronger predictive value of tested risk factors, thus supporting the main results (Appendix B; Table A1, Table A2, Table A3).

## 4. Discussion

In this prospective population-based study we found that low levels of PA in adolescence increased the probability of obesity in young adulthood. Early musculoskeletal pain, in combination with low PA, contributed consistently and additively to the adverse effect. A low PA level in combination with musculoskeletal pain doubled adolescents’ risk of developing obesity by young adulthood, as compared to their physically active peers without musculoskeletal pain. 

These results are in line with findings from prior studies where longitudinal relationships between physical inactivity in adolescence and obesity in young adulthood have been reported [42]. The current study adds to previous knowledge by estimating the probability of obesity in young adulthood given combinations of PA levels and two of the most common and burdensome health challenges among adolescents, musculoskeletal pain and psychological distress [7,8]. Our results indicate that the presence of musculoskeletal pain in adolescence predicts development of obesity by young adulthood. Thus, adolescents’ pain may precede obesity, rather than vice versa. This finding may challenge our current belief that pain is often a consequence of obesity. As an example, increased clinical attention to weight status among adolescents with pain has previously been emphasized based on the high rate of obesity (1/3) found among adolescents with chronic pain who have been consulting pain clinics [24,25]. Thus, the current study adds valuable knowledge to existing research from pediatric pain clinics by demonstrating how musculoskeletal pain reported in a population-based sample of adolescents may contribute to long-term weight-related consequences into young adulthood. 

We further anticipated higher probability of obesity among adolescents struggling with both psychological distress and musculoskeletal pain as the co-occurrence of these health challenges is linked to worse pain and decreased functioning [19]. However, in our study we did not find support for an added effect of early psychological distress on adult obesity when accounting for PA level and musculoskeletal pain. This finding contrasts with results from prior studies that have reported an increased risk of obesity in adulthood among those with depressive symptoms in adolescence [26,43,44]. There is a possibility that pain and withdrawal from social contexts, including regular sports activities, may represent the active agents driving or mediating the previously found adverse effect of psychological distress on development of obesity. A reason for the lack of associations between psychological distress in adolescence and later onset obesity in the present study might, however, also be a lack of statistical power due to the sample size and the relatively few cases of obesity in our material. 

Prospective studies that examine multiple potential risk factors for obesity, as well as their co-occurrence, in diverse samples of adolescents are needed to enhance our understanding of the complexity of factors that may contribute to the development of obesity [32]. Taken together, findings from this study support the need for public health efforts to focus on proactive strategies to promote and customize sports activities for all adolescents. Early identification of pain and subsequent implementation of need-based treatment may represent an opportunity to prevent the development of obesity, as the additional adverse effect of musculoskeletal pain on lower PA levels indicated a strong tendency towards increased probability of obesity. Further, our findings pointed in the same direction for both genders, which strengthens our confidence in the results. Sensitivity analyses, where the reference groups were limited to participants within a normal weight range in young adulthood, revealed even higher probabilities of obesity across combinations of lower levels of PA and occurrence of musculoskeletal pain, and thus confirmed the robustness of the results from our main analysis. 

The results of this study suggest that health care professionals should be aware of the potential increased likelihood of future obesity among adolescents with low levels of PA who also report musculoskeletal pain. In particular, adolescents who are at risk of being obese or having difficulty with weight management may require greater attention in examinations of musculoskeletal pain, assessment of potential underlying causes, and implementation of need-based treatment, in addition to the emphasis on providing information on the benefits of PA and activity guidance.

Of the three exposure factors examined in this study, a low level of PA in adolescence was most strongly associated with obesity in young adulthood. PA is probably the most modifiable of the factors examined, as getting adolescents to be more physically active and to participate in sports and activities may be easier than treating anxiety, depression, and musculoskeletal pain. Additionally, an increase in PA may lead to higher levels of socialization through inclusion in community activities, thereby creating opportunities for development of social skills and social competence, which in turn may provide benefits for musculoskeletal health and mental health [45,46]. Further, based on the current findings, musculoskeletal pain appears to be a stronger predictor than psychological distress. This might be of special importance as musculoskeletal pain could be easier to detect and to modify through interventions than psychological distress. 

Although there is existing evidence to suggest that low levels of PA [11,32], musculoskeletal pain [24,25], and psychological distress [26,27,28] are factors to consider in the prevention of obesity, this study is unique in examining the potential long-term impact of combinations of these adverse health factors on the probability of obesity. The study uses a representative population-based sample of adolescents with an 11-year follow-up period capturing the transition period from adolescence into young adulthood. 

A limitation of our study is that our exposure measures were self-reported. However, PA has been assessed using the WHO HBSC measure of frequency of PA, which has been validated for use in adolescent samples [40,47]. We have used a validated measure to assess psychological distress (SCL-5), with a cut-off point shown to be clinically relevant [41]. The questions about musculoskeletal pain only provided information about pain frequency, not severity, and may be prone to recall bias due to the 12-month recall period. Reports of “neck or shoulder pain” and “joint or muscle pain” were combined into one category of musculoskeletal pain due to the lack of statistical power to analyze them separately, and a drawback may be that information about pain localization is not taken into account. Another limitation is the low follow-up participation rate from the Young-HUNT1 study to the HUNT3 study, which may represent a selection bias. Many young adults had moved out of the county for educational purposes and were not eligible for invitation at follow-up. However, baseline values were comparable between participants and non-participants in the 11-year follow-up from the Young-HUNT1 study to the HUNT3 study regarding PA level and mean BMI [36], indicating that the drop out is unlikely to substantially influence the results. Further, comparisons between these groups showed no significant differences in occurrence of musculoskeletal pain or psychological distress at baseline. 

Although the probability estimates indicated a strong pattern of increased probability of obesity across lower levels of PA in combination with musculoskeletal pain in both genders, the relatively low number of obese young adults resulted in low precision as reflected in the wide 95% CIs, especially for girls where three factors (PA level, musculoskeletal pain, and psychological distress) were included into the final regression model. Hence, we emphasize the patterns in our findings, and recommend caution in the interpretation of the single probability estimates. Further research assessing the combined effects of these health challenges in adolescence on the probability of obesity is needed to confirm the findings, ideally with more frequent follow-ups and improved measurement methods, including more detailed assessment of pain location and severity. 

Lastly, as regression analyses for prediction purposes were conducted to identify subgroups of adolescents at high risk of obesity focusing on the three aforementioned factors, other health risk factors that might make additional contributions to PA limitations and the risk of obesity, such as diet, smoking, alcohol and drug use, have not been taken into account. Further, BMI status at baseline was not considered. However, only 40 adolescents (2%) were obese at baseline.

## 5. Conclusions

The study shows that adolescents with a low PA level may have a higher probability of obesity in young adulthood, and that co-occurrence of musculoskeletal pain further increases the probability of later obesity. Our findings support proactive strategies to make regular sports activities more inclusive and accessible for all adolescents in order to reduce risk of becoming obese in young adulthood. Interventions targeting adolescents’ musculoskeletal pain and underlying causes could make an important contribution in hindering the development of obesity in adolescents at risk.

## Figures and Tables

**Table 1 ijerph-17-04603-t001:** Participants’ characteristics at baseline–nord-trøndelag health study in norway (Young-HUNT1).

Variables	Girls (N = 1049)	Boys (N = 810)	*p*-Value
Age (years), mean (SD)	16.0 (1∙8)	16.0 (1.8)	
Physical activity, N (%)			
High PA	245 (23.4)	239 (29.5)	
Moderate PA	449 (42.8)	321 (39.6)	
Low PA	347 (33.1)	237 (29.3)	0.005 *
Missing	8 (0.8)	13 (1.6)	
Musculoskeletal pain, N (%)			
Yes	379 (36.1)	213 (26.3)	
No	638 (60.8)	568 (70.1)	*p* < 0.001
Missing	32 (3.1)	29 (3.6)	
Psychological distress, (SCL5), N (%)			
SCL5 ≥ 2	123 (11.7)	46 (5.7)	
SCL5 < 2	890 (84.8)	735 (90.7)	*p* < 0.001
Missing	36 (3.4)	29 (3.6)	
BMI, mean (SD)	21.5 (3.2)	21.1 (3.2)	0.012
Obese †, N (%)	16 (1.5)	24 (3.0)	0.027

Boys and girls analysed separately; * Chi-square test for trend; High physical activity (PA) = ≥4 days/week, Moderate PA = 2–3 days/week, Low PA = ≤1 day/week; SCL5 = Hopkins Symptom Checklist Five Items (range 0–4); † BMI-derived categorization of obesity defined by the International Obesity Task Force (IOTF); criteria for adolescents.

**Table 2 ijerph-17-04603-t002:** Results from logistic regression analysis with obesity (BMI ≥ 30) in young adulthood as dependent variable.

Variables	Univariate Analysis			Multiple Analysis		
	OR	95% CI	*p*-Value	OR	95% CI	*p*-Value
**Girls**						
PA level						
Low PA	1.0 (Reference)			1.0 (Reference)		
Moderate PA	0.67	0.46–0.98	0.04	0.67	0.45–0.99	0.05
High PA	0.64	0.41–1.00	0.05	0.61	0.38–0.97	0.04
Musculoskeletal pain						
No	1.0 (Reference)			1.0 (Reference)		
Yes	1.53	1.09–2.16	0.02	1.55	1.08–2.22	0.02
Psychological distress *						
No	1.0 (Reference)			1.0 (Reference)		
Yes	1.41	0.87–2.28	0.17	1.09	0.65–1.82	0.76
**Boys**						
PA level						
Low PA	1.0 (Reference)			1.0 (Reference)		
Moderate PA	0.70	0.46–1.07	0.10	0.67	0.44–1.02	0.06
High PA	0.56	0.35–0.90	0.02	0.55	0∙35–0.89	0.02
Musculoskeletal pain						
No	1.0 (Reference)			1.0 (Reference)		
Yes	1.30	0.88–1.93	0.19	1.30	0.87–1.93	0.20
Psychological distress *						
No	1.0 (Reference)			1.0 (Reference)		
Yes	1.12	0.53–2.38	0.77	**-**		**-**

High PA = ≥4 days/week, Moderate PA = 2–3 days/week, Low PA = ≤1 day/week; * SCL-5 ≥ 2, SCL5 = Hopkins Symptom Checklist Five Items (range 0–4).

**Table 3 ijerph-17-04603-t003:** Risk matrix model for girls. Probability of obesity (BMI ≥ 30 kg/m^2^) (percentage, (95% confidence interval (CI)) in young adulthood.

		PA Level		
		Low PA	Moderate PA	High PA
**MS pain**	**Psychol. distress**	25%	18%	17%
		(11–39%)	(3–33%)	(1–33%)
	**No psychol. distress**	24%	17%	16%
		(15–33%)	(10–24%)	(8–24%)
**No MS pain**	**Psychol. distress**	18%	13%	12%
		(1–35%)	(0–32%)	(0–35%)
	**No psychol. distress**	17%	12%	11%
		(11–23%)	(8–16%)	(6–16%)

Physical activity (PA) level (low PA = ≤1 day/week, moderate PA = 2–3 days/week, high PA = ≥4 days/week), presence of musculoskeletal (MS) pain (never/seldom or sometimes/often) and psychological distress (SCL5, <2 points or ≥2 points). Red = highest risk profile.

**Table 4 ijerph-17-04603-t004:** Risk matrix model for boys. Probability of obesity (BMI ≥ 30 kg/m^2^) (percentage, (95% CI)) in young adulthood.

	PA Level		
	Low PA	Moderate PA	High PA
**MS pain**	27%	20%	17%
	(16–38%)	(12–28%)	(7–27%)
**No MS pain**	22%	16%	14%
	(16–28%)	(11–21%)	(9–19%)

Physical activity (PA) level (low PA = ≤1 day/week, moderate PA = 2–3 days/week, high PA = ≥4 days/week), presence of musculoskeletal (MS) pain (never/seldom or sometimes/often) and psychological distress (SCL5, <2 points or ≥2 points). Red = highest risk profile.

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
