# Peer review of "Obesity in Young Adulthood: The Role of Physical Activity Level, Musculoskeletal Pain, and Psychological Distress in Adolescence (The HUNT-Study)"

_ijerph, 2020, doi:10.3390/ijerph17124603_

Round 1

Reviewer 1 Report

Dear authors,

Their work is well structured and documented, for which I congratulate them. However, I mase some comments so that you can assess them:

  • Title: It is too long. I don't know if you could reduce it by keeping the keywords.
  • Methods: It would be favourable if you could include the socio-economic and cultural characteristic data of the people investigated. It would also be positive to know other variables, such as the diet of the people studied.

Best regards. 

Author Response

Point 1: Title: It is too long. I don't know if you could reduce it by keeping the keywords.

Response 1: Thank you for making that point, we recognize that the title is too long. According to the guidelines from the HUNT Research Centre, “The HUNT Study” should be included in the title. We have put this study name in brackets, and have excluded the information about the prospective design, page 1: “Obesity in young adulthood; the role of physical activity level, musculoskeletal pain, and psychological distress in adolescence (the HUNT-study)”.

Point 2: Methods: It would be favourable if you could include the socio-economic and cultural characteristic data of the people investigated. It would also be positive to know other variables, such as the diet of the people studied.

Response 2: Thank you for the suggestions. We agree that including socio-economic and cultural characteristics of our study sample would be informative. Unfortunately, participants in Y-HUNT1 were not asked about socio-economic conditions. Thus, we are not able to elaborate on these baseline characteristics. Participants were asked about meals and eating habits, but currently we do not have access to these data. This would have required a new application specifically for the use of these variables to the HUNT Research Centre.

We have, however, added relevant information about the cohort profile of the HUNT study, as well as information from Statistics Norway, in Materials and Methods (2.1 Study sample) page 3, lines 121-125:

The majority of the population in the Nord-Trøndelag County is Caucasian. In terms of sex, age distribution, mortality and health status, the population is representative of Norway (Holmen et al., 2014, Krokstad et al., 2013), and the development of obesity in Nord-Trøndelag is comparable to the global trend (Ng et al., 2014). The Norwegian society is a socioeconomically stable society with a high degree of equity among students and young people (Statistics Norway).

Further, not considering diet has now been mentioned as a limitation in the discussion, page 11, line 401.

Reviewer 2 Report

I would like to congratulate the authors as it is not easy to conduct an 11-year prospective study. The study is in general well written and the outcomes add valuable knowledge. However, there are significant flaws in the introduction, methods and results that have to be addressed before considering this manuscript for publication. Please, see my comments attached.

INTRODUCTION

Introduction in general

I think the authors failed in:

  • Explaining the relationship between the three factors they talk about.
  • Identifying the gap in the literature, what has been already done? what do we know about this topic? Why the three factors are important? The authors should address these questions in order to provide an appropriate insight to the reader.

Third paragraph: I do not see the link of this paragraph with the first two paragraphs

  • Line 55-56: the authors state that musculoskeletal pain and mental health are major public health problems. But they state the same for obesity in the first paragraph. Therefore, I somehow miss the point. Maybe the authors should introduce obesity, musculoskeletal pain, and mental health together in the first paragraph and explain the connection between these three factors.
  • Line 55 to 56: I think that a reference might be required to support this statement
  • Line 56 to 57: “these problems commonly co-occur” So how much these problems co-occur, who is affected, what is the prevalence of these factors? I think this statement needs further clarification.
  • Line 57: “and are known to be leading causes of health-related disability,” I think the authors might add one or two health-related disability, so the reader understands what the authors mean
  • All paragraph: I have been unable to understand the point of this paragraph because the authors go back to obesity problems instead of explaining the connection between obesity and musculoskeletal pain and mental health problems

Lines 64 to 70: I think several references are required in this paragraph. Also, I think the authors have failed in explaining the relationship between obesity, musculoskeletal pain and mental health problems. So, I do not see why they should be targeted together.

METHODS

  • Line 81 to 82: I recommend the authors to change “(10 202 residents)” for (n = 10 202). I think readers might misunderstand the word “residents” and think the authors mean all the population in Nord-Trøndelag Health instead of the adolescents between 13-19
  • Line 81: Why the authors included adolescents over 18 years old?
  • Line 84: is the health-related questionnaire different to the questionnaire describes in the subsection “2.3. Exposures”? If so, the authors should provide further information. If not, the authors should explain that the questionnaire is described somewhere else in the manuscript
  • line 91: same question. Is the health health-related questionnaire different from the questionnaire describes in the subsection “2.3. Exposures”? if so, is it different from the questionnaire that the authors mean in line 84?
  • HUNT1 and HUNT3: I understand that those people who did not participate in HUNT 1 were excluded, I am right? Therefore, I do not understand the lines 88-93. I think the authors should merge the first and second paragraph of Methods (lines 81-93) and only focus on those subject that participated in this study or were eligible for this study.
  • Line 97 to 98: Why people classified as underweight were excluded?

Data analysis

  • Did the authors test the normality of the data?
  • Line 149: Why p-values ≤0.2 was selected as a cut off for the multiple logistic regression analysis?
  • Line 161: In line 94 to 98 the authors somehow are stating that those who participated in HUNT1 and HUNT3 were selected for the study. Therefore, I do not understand the “… and those who did not participate in HUNT3,”

RESULTS

Table 1:

  • In tables, I always recommend using as lower lines as possible. In fact, usually, only three horizontal lines are used. It usually looks more professional.
  • Why is p-value only reported for Low PA. Why it is not reported for High PA and Moderate PA? Why did the authors do not include Z-scores?
  • Regarding musculoskeletal pain, why is p-value only reported for NO?7
  • Regarding Psychological distress, why p-value is only reported for SCL5 <2?

** In Table 1 the authors are comparing between boys and girls. This fact has to be explained in Table’s foot.

Table 2:

  • Once again, I encourage the authors to use the fewer number of lines as possible

Figures 1 and 2:

  • I think they are not figures, but tables. So, I recommend the authors to treat them as tables.
  • I recommend the authors not to use colours unless they are strictly necessary. 1) because it might increase the publishing cost; 2) because it makes hard to read to colour blind readers. The authors might use a symbol instead of colours.
  • Once again, I encourage the authors to use the fewer number of lines as possible

Discussion

  • Line 264 to 265: recommend the authors to quote the two burdensome health challenges that are explored in this study
  • Line 299 to 304. I think that some references might be required to support some of the statements of this paragraph.
  • Lines 309 to 311. I think a reference is required to support this statement.
  • Lines 315 to 316. I think a reference is required to support this statement.

Author Response

INTRODUCTION

Point 1: Introduction in general. I think the authors failed in: Explaining the relationship between the three factors they talk about. Identifying the gap in the literature, what has been already done? what do we know about this topic? Why the three factors are important? The authors should address these questions in order to provide an appropriate insight to the reader. Third paragraph: I do not see the link of this paragraph with the first two paragraphs.

Response 1: Thank you for the constructive feedback and for the suggestions to improve the introduction. We recognize the need to provide readers with more in-depth insight into the gap in the literature and to re-arrange the introduction.

After introducing the outcome obesity as a major public health concern and highlighting the need for preventive efforts, we have introduced all three exposures of interest together in the second paragraph, in the Introduction, page 2, lines 46-49:

“A lifestyle characterized by low levels of PA, musculoskeletal pain, and psychological distress are three common and related factors adversely affecting the health and well-being of adolescents worldwide (Guthold et al., 2020, Kamper et al., 2016, Kieling et al., 2011, Bertha et al., 2013, Mokdad et al., 2016). The cumulative presence of these factors could be of particular importance for the development of overweight and obesity over time”.

We have further focused on why these factors are important, and have provided information about rates of insufficiently active adolescents, as well as estimated prevalence rates for musculoskeletal pain and mental health problems, page 2, lines 51-53:

Young people commonly drift away from engagement in regular physical activity throughout their adolescent years (Dumith et al. 2011), and the majority of adolescents worldwide (81%) do not adhere to the current recommendations of daily PA (Gothold et al.,2020).

Page 2, lines 62-64:

Estimations indicate that between one third and half of adolescents worldwide report musculoskeletal pain monthly or more (Kamper et al., 2016)”, and up to one in five children and adolescent experience mental health problems (Kieling et al., 2011, Belfer et al., 2008)”.

Further, a more comprehensive description of previous studies examining relationships between 1) musculoskeletal pain and 2) psychological distress and the outcome obesity has been added in the Introduction, page 2, lines 72-78:

“The potential adverse impact of pain on development of obesity has been raised as a concern based on the high prevalence of obesity seen in adolescents with chronic pain (Wilson et al.,2010, Santos et al., 2017). Further, results from systematic reviews have found support for psychological distress or depression/depressive symptoms in adolescence being predictive for development of obesity (Blaine 2008, Luppino et al., 2010, Mannan et al., 2016, Vamosi, Heitmann & Kyvik, 2010). Limited evidence, however, exists on individual and combined effects of these common health problems from population-based cohorts with follow-up into young adulthood”.

Point 2: Line 55-56: The authors state that musculoskeletal pain and mental health are major public health problems. But they state the same for obesity in the first paragraph. Therefore, I somehow miss the point. Maybe the authors should introduce obesity, musculoskeletal pain, and mental health together in the first paragraph and explain the connection between these three factors.

Response 2: Thank you for pointing out this lack of clarity. As outlined in the response above, we have introduced obesity, the outcome of interest, before introducing our focus on three major adverse health factors in adolescence (physical activity level, musculoskeletal pain and psychological distress) potentially impacting obesity in young adulthood. Now, we have focused more on explaining the extent of these health problems, as well as added information about the current gap in the literature and the need for studies to assess multiple factors, and their co-occurrence, associated with future obesity.

Point 3: Line 55 to 56: I think that a reference might be required to support this statement.

Response 3: Thank you for reminding us to include references to support this statement, page 2, line 60-62:

“Further, musculoskeletal pain and mental health problems frequently takes place in adolescence, and constitute among the top causes of functional impairments in adolescence (Kamper et al., 2016, Kessler et al., 2005, Kieling et al., 2011, Mokdad et al., 2016)”.

Point 4: Line 56 to 57: “these problems commonly co-occur” So how much these problems co-occur, who is affected, what is the prevalence of these factors? I think this statement needs further clarification.

Response 4: We have added information about the prevalence of these factors in the introduction, page 2, lines 62-64:

“Estimations indicate that between one third and half of adolescents worldwide report musculoskeletal pain monthly or more (Kamper et al.,2016), and up to one in five children and adolescent experience mental health problems (Kieling et al., 2011; Belfer et al., 2008)”.

Further, co-occurrence rate of these health problems has been added, lines 64-66:

“In a national representative cohort from the United States, 26% of adolescents reported that they had experienced both chronic pain and mental health problems during their lifetime (Tagethoff et al., 2015.

Point 5: Line 57: “and are known to be leading causes of health-related disability,” I think the authors might add one or two health-related disability, so the reader understands what the authors mean.

Response 5: We agree that this should be stated. Examples of health-related disabilities has been added, page 2, lines 66-70:

Both musculoskeletal pain and mental health problems are known to be leading causes of health-related disability (GBD Compare, 2017, Kieling et al., 2011, Mokdad et al., 2016), and are among adolescents associated with impairments in daily activities, social relationships and extracurricular activities, as well as sleep disturbances and absence from school (Roth-Isigkeit et al., 2005, Kamper et al., 2016, Soltani et al., 2019)”.

Point 6: All paragraph: I have been unable to understand the point of this paragraph because the authors go back to obesity problems instead of explaining the connection between obesity and musculoskeletal pain and mental health problems.

Response 6: We hope that this has been clarified in our responses above, and in our changes and additions to the Introduction. We have provided information about the evidence from previous studies investigating the potential impact of adolescent PA, musculoskeletal pain and mental health problems on obesity in adulthood. Further, the need for research effort into examining how multiple factors and their co-occurrence may affect adult obesity has been emphasized, as well as the lack of studies covering the transition period from adolescence to young adulthood.

Point 7: Lines 64 to 70: I think several references are required in this paragraph. Also, I think the authors have failed in explaining the relationship between obesity, musculoskeletal pain and mental health problems. So, I do not see why they should be targeted together.

Response 7:  We agree, and references have been added, page 2, line 86:

Early intervention is a key in the prevention of obesity (Weihrauch-Blüher et al.,2018, Street, Wella & Hills, 2015, Doak et al.,2006),…”.

Lines 88-91: Physical activity habits, as well as musculoskeletal pain and mental health challenges, are likely to be less ingrained in adolescence than in adulthood, emphasizing the importance of preventive efforts during the adolescent years when these potential risk factors are more malleable (Patton et al., 2016)”.

Reasons for examination of the co-occurrence of PA, musculoskeletal pain and mental health problems has been outlined in the Introduction, as elaborated on in the responses above (point 1,2 and 6).

METHODS

Point 8: Line 81 to 82: I recommend the authors to change “(10 202 residents)” for (n = 10 202). I think readers might misunderstand the word “residents” and think the authors mean all the population in Nord-Trøndelag Health instead of the adolescents between 13-19

Response 8: Thank you for this remark. Changes have been made accordingly page 3, line 104.

Point 9: Line 81: Why the authors included adolescents over 18 years old?

Response 9: We included adolescents over 18 years old, according to the WHO definition of 'adolescents' as individuals between 10-19 years of age. Individuals ≥20 years of age were excluded.

Point 10: Is the health-related questionnaire different to the questionnaire describes in the subsection “2.3. Exposures”? If so, the authors should provide further information. If not, the authors should explain that the questionnaire is described somewhere else in the manuscript

Response 10: Thank you for pointing out this lack of information. The health-related questionnaire responded by the adolescents contains self-reported physical activity, occurrence of musculoskeletal pain and psychological distress, further described in the subsection “2.3. Exposures”. Information has been added on page 3, lines. 106-108:

“During school hours, the adolescents completed a comprehensive health-related questionnaire about lifestyle factors, health, and quality of life, including the exposure variables described below (2.3. Exposures)”.

Point 11: Line 91: same question. Is the health health-related questionnaire different from the questionnaire describes in the subsection “2.3. Exposures”? if so, is it different from the questionnaire that the authors mean in line 84?

Response 11: The adult self-reported health-related questionnaire differs somewhat from that given to the adolescent. However, in this study, responses to the adult questionnaire were not part of the exposures nor outcomes, as only objectively measured BMI was used to define the outcome obesity (BMI ≥30 kg/m2) in young adulthood.

Point 12: HUNT1 and HUNT3: I understand that those people who did not participate in HUNT 1 were excluded, I am right? Therefore, I do not understand the lines 88-93. I think the authors should merge the first and second paragraph of Methods (lines 81-93) and only focus on those subject that participated in this study or were eligible for this study.

“These adolescents were also invited as young adults 11 years later, in 2006-08, to participate in the HUNT3 study”.

Point 13: Line 97 to 98: Why people classified as underweight were excluded?

Response 13: Only 1,5% (n=29) in our material were classified as underweight, as described in 2.1 Study sample and in the flow chart (Appendix A). As these individuals are considered at higher risks of serious illness, they were excluded from the analysis.

Point 14: Did the authors test the normality of the data?

Response 14: Exposure variables are ordinal (PA-level) and dichotomized (MS pain and psychological distress). The outcome measure, Obesity, is dichotomized (BMI ≥30 kg/m2). As the dependent variable has two levels, we have used a binary logistic regression model, estimating odds ratios. Thus, we did not use linear regression analyses which would require the residuals to follow a normal distribution. Concerning our descriptive analyses, only age was normally distributed and therefore described with mean and standard deviation (SD). We have plotted the variable and performed a visual inspection of the histogram and the q-q plot and concluded that age was normally distributed.

Point 15: Line 149: Why p-values ≤0.2 was selected as a cut off for the multiple logistic regression analysis?

Response 15: Thank you for raising an interesting question. There are no clear rules as to which cut-off values for p-values from univariate analyses to use when including variables into multiple models. A rule of thumb is either using p<0.1 or p<0.2. In our model, the main aim was to develop a prediction model, not to reveal variables which would be independently associated with the outcome. Thus, we wanted to include all variables that would improve the prediction ability of our model. Moreover, some variables might be statistically significant only when other variables are included in the multiple model, so we did not want to rule out such possible interactions.

Point 16: Line 161: In line 94 to 98 the authors somehow are stating that those who participated in HUNT1 and HUNT3 were selected for the study. Therefore, I do not understand the “… and those who did not participate in HUNT3,”

Response 16: Yes, those who participated in both Young-HUNT1 and HUNT3 (11-years follow-up) were selected for this study. However, as non-participation in the follow-up may be a potential source of selection bias, we included information about baseline characteristics between Young-HUNT1 participants who also participated in the follow-up in HUNT3 (and thereby were selected into this study) and those who did not participate in in the follow-up. Further, as describing such efforts to address potential sources of bias is recommended in the STROBE checklist of observational studies, we chose to include this information both in the Method section and in the discussion of limitations. Nevertheless, if it is desirable to exclude this issue in order to avoid misunderstandings, we can accept that this information will be removed from the article.

RESULTS

Point 17: Table 1:In tables, I always recommend using as lower lines as possible. In fact, usually, only three horizontal lines are used. It usually looks more professional.

Response 17: We agree, and have formatted the tables accordingly.

Point 18: Table 1: Why is p-value only reported for Low PA. Why it is not reported for High PA and Moderate PA? Why did the authors do not include Z-scores?. Regarding musculoskeletal pain, why is p-value only reported for NO? Regarding Psychological distress, why p-value is only reported for SCL5 <2?

Response 18: PA level is an ordinal variable with three categories, and we have therefore inserted the p-value from the Chi-square test for trend in Table 1. As the variables for musculoskeletal pain and psychological distress are dichotomized, only one p-value are given for each of these variables. We have performed many statistical tests so tested for possible differences on a global level and did not add any additional tests comparing groups for selected categories of presented variables. For example, here it would be proportions of low PA, proportions of High PA – we have only included tests for trend to minimize multiple testing. We present all p-values with 3 decimals, thus with high level of precision. We did not find it informative to include the actual test statistics (z-scores) as they are used to derive p-values, which are presented.

Point 19: ** In Table 1 the authors are comparing between boys and girls. This fact has to be explained in Table’s foot.

Response 19: We have added information on separate analyses for boys and girls in the Table’s foot, page 6, line 222.

Point 20: Table 2: Once again, I encourage the authors to use the fewer number of lines as possible

Response 20: We have formatted Table 2 (page 8) and Table A1 (Appendix B, page 14) accordingly.

Point 21: Figures 1 and 2: I think they are not figures, but tables. So, I recommend the authors to treat them as tables. I recommend the authors not to use colours unless they are strictly necessary. 1) because it might increase the publishing cost; 2) because it makes hard to read to colour blind readers. The authors might use a symbol instead of colours. Once again, I encourage the authors to use the fewer number of lines as possible

Response 21: Thank you for making that point. As recommended, the risk matrix models are now presented as tables. It is strictly not necessary to use colours, although we think this would help to visualize the pattern of the findings. Based on your comment, we do however suggest to only highlight the highest risk profile, as this will be apparent also to colour blind readers. Changes are made, page 9 and page 15.

DISCUSSION

Point 22: Line 264 to 265: recommend the authors to quote the two burdensome health challenges that are explored in this study

Response 22: These have been quoted on page 10, line 315.

Point 23: Line 299 to 304. I think that some references might be required to support some of the statements of this paragraph.

Point 24: Lines 309 to 311. I think a reference is required to support this statement.

Response 24: We agree. References to support this statement have been included, page 11, lines 363-364.

Point 25: Lines 315 to 316. I think a reference is required to support this statement.

Response 25: References to support this statement have been included, page 11, lines 367-368.

Round 2

Reviewer 2 Report

I would like to congratulate the authors for the excellent work they have addressed. I especially acknowledge the difficulty of developing longitudinal studies. On the other hand, in my opinion, the authors have effectively addressed all the requested changes. Moreover, the quality of the manuscript is in line with the standards of the IJERPH journal and the interest of the readers. However, before publishing the manuscript I think the authors might address the minor changes I identify below.

INTRODUCTION

Line 64. I would add “moreover” before “These problems commonly co-occur”

METHODS

Line 117. I think the authors should explain in the main text that the participants ≥ 20 were excluded following the criteria of WHO to define adults. Thus, readers will know the reason.

Line 119. Same for underweight. I liked the response to my question in my previous revision. So, I just recommend the authors to add the explanation in the main text

Data Analysis

I think the authors should explain the p-value for the multiple logistic regression analysis was set at p ≤ 2. I liked the response to my question in my previous revision. So, I just recommend the authors to add the explanation in the main text

Author Response

Response to Reviewer 2 Comments

Point 1: Line 64. I would add “moreover” before “These problems commonly co-occur”

We agree and have added “moreover” in line 64.

Response 1: We agree and have added “moreover” in line 64.

Point 2: Line 117. I think the authors should explain in the main text that the participants ≥ 20 were excluded following the criteria of WHO to define adults. Thus, readers will know the reason.

Response 2: This explanation has been added, line 117-118.

Point 3: Line 119. Same for underweight. I liked the response to my question in my previous revision. So, I just recommend the authors to add the explanation in the main text.

Response 3: This explanation has been added, lines 120.

Point 4: Data Analysis. I think the authors should explain the p-value for the multiple logistic regression analysis was set at p ≤ 2. I liked the response to my question in my previous revision. So, I just recommend the authors to add the explanation in the main text

Response 4: A more detailed explanation have has been added in to 2.5 Data analyses, line 177-182.
